# Performance Feedback, Goal Clarity, and Public Employees' Performance in Public Organizations

**Soonae Park [1] and Sungjoo Choi [2],***

[1]   Graduate School of Public Administration, Seoul National University, Seoul 08826, Korea; psoonae@snu.ac.kr
[2]   Department of Public Administration, Kyung Hee University, Seoul 02447, Korea
*   Correspondence: sungjoochoi@khu.ac.kr

**Abstract:** Scholars have emphasized the importance of supervisory feedback in improving individual performance. Subordinates benefit from clear communication of organizational goals and expected behaviors of employees, which are linked to the improvement of individual performance and organizational effectiveness. We examine the dynamic relationship between feedback on performance and individual performance, which is mediated by performance goal clarity. Given the potential influence of contextual factors on the relationship, we also test the moderation effect of autonomy on the relationship between performance goal clarity and individual performance. The data collected from the local government workforce in Korea were analyzed through structural equation modeling. The findings show that performance feedback is significantly and positively related to individual performance, mediated by performance goal clarity. In addition, the mediation effect of performance goal clarity was positively moderated by autonomy. The results imply that performance feedback can contribute to the improvement of individual performance by helping employees clearly understand the performance goals they need to accomplish. The higher levels of autonomy may promote the positive link between a clear understanding of performance goals and individual performance.

**Keywords:** feedback; goal clarity; and performance

## 1. Introduction

Managing organizational performance is directly related to organizational sustainability. Sustainable organizations adopt strategies and activities that are accountable for the demands of the organization and its stakeholders, while also protecting, maintaining, and increasing the human and financial resources that the organization will need in the future [1]. The enhanced organizational performance in responding to external demands and managing internal resources may offer organizations a higher probability of sustainability. Public organizations are not exceptions. Over the past several decades, a large volume of literature in public management has delved into strategies to enhance the performance of public organizations and has demonstrated the critical role of management in leading to higher performance [2,3].

Managing performance in public organizations, however, has been quite challenging due to the complex nature of organizational goals. The goals of public organizations are inclined to be more ambiguous, dynamic, and sometimes multifaceted than those in private organizations. [4] Operating in highly political environments, public organizations have often struggled to pursue multiple values (e.g., equity, efficiency, democratic values) to cope with competing goals and to reduce goal ambiguity [5–8]. Under such circumstances, managerial strategies and efforts to help employees select among competing goals and prioritize between them will be necessary [7].

Scholars [6,8] have suggested that performance feedback may alleviate the negative effects of low goal clarity on performance in public organizations by guiding employees to focus on selected goals

by decision-makers in the organization. They highlighted the importance of enhanced interactions between supervisor and subordinate, including clear communication of organizational goals and expected behaviors of employees and sharing performance information, which are likely to aid employees in accomplishing higher performance [7,9]. Thus, performance feedback may play a more crucial role in managing performance in the context of public organizations, which have suffered from unclear organizational goals, than in any other organizational setting.

Research that has shown the positive effects of performance feedback on organizational effectiveness is not rare. Quite a few studies in business management have demonstrated the positive effects of performance feedback on organizational effectiveness including individual and organizational performance [10–15]. However, relatively less research has explored how active utilization of performance feedback can help public organizations and their employees enhance performance. Given a highly politicized and complex environment where public organizations operate, the external validity of the findings from private businesses may be questionable, requiring further investigation.

The purpose of this research is twofold. First, we examine if performance feedback will contribute to employees' performance in the context of public organizations. By analyzing the data collected from public employees in local governments in Korea, we test if the positive link between performance feedback and employees' performance is also found in public organizations. Structural equation modeling was adopted to estimate the hypothesized relationship between performance feedback and individual performance. Second, we investigate the process of performance feedback affecting individual performance. In particular, we focus on the mediating role of goal clarity between performance feedback and individual performance, assuming that performance feedback will clarify performance goals and desirable behavioral standards for employees, and eventually assist them in improving their performance. Given that public organizations have suffered from lower levels of goal clarity, we expect that performance feedback will draw critical attention from public managers as a strategy to solve this chronic problem of public organizations [7,16].

First, we review the relevant literature. Grounded upon the literature review, the hypotheses will be developed. Next, the data will be statistically analyzed to test the hypotheses. Finally, the results and implications will be discussed.

## 2. Theoretical Framework

### 2.1. The Performance Feedback Effects

Performance feedback refers to "information about the actual performance or actions of a system used to control the future actions of a system" [11,17] p. 310. It has some advantages, such as cost-effectiveness, programmatic simplicity and flexibility, and an emphasis on positive consequences, and therefore adopted as an organizational intervention technique to enhance performance [18] p. 3. It is less likely to use aversive control procedures by weighing more on positive outcomes [18]. Performance feedback could be offered in various ways. Scholars have tested the effectiveness of offering performance feedback in a positive and negative way [10,15]. Positive performance feedbacks are favorable comments or appreciation expressed by supervisors to subordinates through sharing performance information, whereas negative performance feedbacks are negative performance information and criticisms from supervisors [15,19]. Positive performance feedbacks serve as the reinforcer of desirable behaviors contributing to individual productivity and professional development, but negative performance feedbacks could cause subordinates' negative psychological consequences, including a feeling of frustration and decreased self-efficacy [10,15]. Empirical evidence has consistently shown that positive performance feedbacks are effective in promoting individual performance [10,15]. The effectiveness of negative performance feedbacks is inconclusive. Some found that a negative way of delivering performance feedback failed to bring higher performance, whereas others demonstrated

that both positive and negative performance feedback effectively helps to enhance performance only if supervisors focus on providing performance information and deliver it consistently [9,15,20].

Previous studies have investigated the multiple functions of performance feedback are various antecedents leading to performance [10,21,22]. Accurate feedback from a supervisor can yield a number of positive results for subordinates, for example, a better understanding of organizational goals and expected roles and levels of performance and information about job tasks that can facilitate performance [9]. Performance information received from supervisors could be used as a developmental tool that aids employees in modifying their efforts and behaviors to remedy performance deficits or to reinforce desirable behaviors and attitudes producing higher performance [15,19]. Feedback could also generate an instrumental motive that encourages employees to seek for the perceived feedback as well as to self-regulate based on the feedback [14]. We, however, limit our discussion to the potential function of performance feedback that clarifies organizational objectives and performance goals for employees, which will, in turn, promote individual performance and organizational effectiveness.

## 2.2. Hypotheses: Performance Feedbacks and Performance

Goal-setting theory and control theory offers theoretical grounds for postulating employees who can benefit from performance feedback ultimately produce higher performance. Goal-setting theory [23] noted that in the process of accomplishing the goals, feedback plays a guiding role in directing individual workers to follow the behavioral standards and expectations and to pay attention to the aspects of tasks indicated by feedback. In consequence, performance feedback can lead individuals' future goal setting and behaviors to the direction of promoting their productivity, contributing to higher performance of the organization [19,24]. In a similar vein, control theory explained that performance feedback reduces the gap between the current level of performance of an individual and the expected standards set by the organization [25]. In case employees' goals are not congruent with those of the broader organization, the organization may not benefit from the contribution of individual workers, which will not be incorporated with the organization's needs. Individuals can attain goals and outcomes valued by the organization through the process of adjusting their understanding of the goals and expected behaviors to the established standards by following the feedback.

Many studies have convincingly demonstrated the positive connection between feedback and performance under various circumstances [2,10,12,13,15]. Favero et al. (2016) examined how internal management efforts including performance feedback provision affect school performance. The results were consistent with the literature in public management, showing that managerial efforts are effective in improving organizational performance. Su et al. (2019) found a positive link between developmental feedback and employee performance with evidence of the impacts of contextual factors on the relationship. The relationship between performance feedback and performance was partially mediated by feedback-seeking behaviors. Employees with political skills were more likely to request performance feedback from their supervisors and improve job performance. Similarly, Guo et al. (2014) found that developmental feedbacks are positively associated with job performance. Intrinsic motivation partially mediated the relationship between feedback and job performance. The method of delivering the performance feedbacks also seem to affect the effectiveness of performance feedback. Zheng et al. (2013) examined the relationship between positive and negative performance feedback and task performance. It was observed that only the positive way of offering performance feedback was positively related to employee task performance. Negative performance feedback, although not significantly associated with task performance, reinforced the effects of positive performance feedback on performance. Contrarily, Choi et al. (2018) conducted a research experiment with participants consisting of students from a university to compare the effects of different types and sequences of providing performance feedback. The results showed that both positive and negative performance feedbacks were effective in enhancing work performance. The positive effects were greater when the way of delivering performance feedbacks was consistent (positive–positive or negative–negative).

Along this argument, we assume that performance feedback can contribute to the improvement of individual performance.

**Hypothesis 1 (H1).** *Performance feedback will be positively related to individual performance.*

*2.3. The Contextual Influences: Performance Goal Clarity and Autonomy*

The generic theoretical discussion underscores the positive effects of performance feedback on task performance [26]. However, empirical evidence does not always seem to support it [14,27–29]. According to relevant meta-analysis, the relationship between feedback and performance is equivocal, generating inconsistent research findings [24]. Scholars suspect that it might be because the relationship between feedback and performance is complex and possibly indirect, affected by various contextual factors [9,14,29]. It thus requires a more sophisticated approach to delve into the dynamic relationships between performance feedback, contextual factors, and individual performance.

Whitaker and his colleagues (2007) explained two potential reasons. First, factors that can potentially mediate the link between feedback and performance may exist. For example, according to Morrison's model of employee information seeking, individuals tend to seek feedback to reduce uncertainty in the work process and increase job knowledge linked to higher performance. Reduced uncertainty, then, leads to desirable work attitudes and higher performance. Similarly, Taylor et al. (1984) noted that employees' clear understanding of behavioral standards through feedback will result in positive changes in performance [30]. The arguments are also consistent with the logic offered by the goal-setting theory discussed earlier. Second, feedback from different sources may lead to different results. Renn and Fedor (2001) noted that feedback-seeking from the supervisor and coworkers may affect the link between feedback and task performance differently [31]. For example, employees are more likely to seek sufficient and relevant feedback from a supervisor than coworkers, which will have greater positive impacts on job clarity and performance.

We, thus, test the potential mediation effect of performance goal clarity between feedback and individual performance. We also investigate the moderating effect of autonomy, which has often been discussed as an important antecedent of higher performance, on the relationship between performance goal clarity and individual performance.

2.3.1. The Mediating Effect of Performance Goal Clarity

Performance goal clarity has often been discussed as a mediating factor that intervenes in the relationship between performance feedback and performance. Goal-setting theory suggests that a clear understanding of performance goals through specific guidance will yield higher performance than merely encouraging employees "to do their best" and not offering a clear direction toward goals and expected behaviors [23,32]. Along the similar line, because organizational goal clarity plays a directing role in channeling and concentrating team motivation to the attainment of the goal, work teams will intensify their efforts toward the goals and accomplish them in more effective ways [11]. Organizational goal clarity can also help work teams envision desirable behaviors, which can contribute to the organization and attain the knowledge of the goals valued by the organization [11,33,34].

Empirical studies consistently found that a clear understanding of performance goals and roles assigned to an individual mediates the relationship between feedback and performance. Whitaker and his colleagues (2007) argued that the seemingly inconsistent relationship between performance feedback and performance may be understood from the perspective of role clarity, which possibly mediates the relationship between feedback and performance. They, indeed, found the mediation effect of role clarity on the relationship between a feedback-supportive environment and an individual's performance. Gonzalez-Mule et al. (2016) also demonstrated that feedback coupled with greater team autonomy may enhance team performance through clarifying the organization's goals and communicating performance information for work teams. Anderson and Stritch (2015), through a laboratory experiment, have shown that individuals who were provided a clear direction of task goals

were able to perform better than others who were not. Based upon these arguments, we predict that performance goal clarity will mediate the relationship between performance feedback and performance where more feedback will improve individual performance through clarifying performance goals.

**Hypothesis 2 (H2).** *Performance goal clarity will positively mediate the relationship between performance feedback and individual performance.*

### 2.3.2. The Moderating Effect of Autonomy

Previous research has suggested that various contextual factors may moderate the relationship between performance goal clarity and performance [11,14,32,35]. Anderson and Stritch (2015), in their experiment, found that task significance affects the association of task goal clarity and performance in the way that when an individual perceives significance of the task, he or she is likely to feel performance pressure and anxiety, which will reduce individual performance. Wallace and his colleagues (2011) found that employees' autonomous power is likely to bring higher performance only when they feel higher accountability for their work. Even in regard to affectional outcomes (e.g., job satisfaction), job autonomy is inclined to affect the outcome conditional to other contextual factors (job demand and goal ambiguity). Jong (2016) reported that job autonomy is likely to increase job satisfaction of individuals, interacting with job demand and goal ambiguity.

In particular, we focus on employees' work autonomy, which, coupled with performance goal clarity, can boost its positive effect on performance. We assume that the synergic effect of autonomy and performance goal clarity will contribute to individual performance. Prior research has often discussed the positive impacts of work autonomy on performance and work attitudes (e.g., job satisfaction, organizational commitment, job involvement) [36–40]. Autonomy, as an internal cognitive state, which can be gained through sharing power and is involved in decision making, leads to increased intrinsic work motivation and enhanced self-efficacy [38,41,42]. Autonomous employees are expected to produce higher performance through sharing performance information, job-related knowledge, and discretion over task, even in highly turbulent work environments [36,39].

Higher autonomy provides individuals with the ability to determine what goals they should pursue for their organization to carry out higher performance and calibrate their efforts toward the organization's goals and individual goal accomplishment [11]. Although individuals with higher autonomy are motivated to make voluntary efforts towards goal attainment, there is no guarantee that they are well aware of the organization' goals, will select the goals consistent with those of the organization, and take a series of actions beneficial to the organization. Some scholars warned that autonomy may put the organization in a risk of disorder when autonomous work teams or individuals pursue goals that are not congruent with those of the organization [11,43,44]. Thus, as goal setting theory notes, clear understanding of goals, which can help employees concentrate their effort on meeting the organization's expectations over them, will be necessary to enhance the benefits of work autonomy [23]. Indeed, some practical experiment is supportive of the potential interaction of autonomy and performance goal clarity. The Texas Instrument company did an experiment on employees for the purpose of designing autonomous work groups. After announcing that employees are autonomous and allowed to do what they want to do, the management encouraged employees to direct themselves and independently act [45]. The results were not desirable because employees, who were not provided the direction of what are the organization's expectations over them, did not know what to do. However, once the organization provided feedback on the goals and goal processes for work groups, highly autonomous work groups started to produce desirable outcomes for the organization, exercising significant levels of authority over work processes and decision making related to tasks [11]. Along with the line of arguments, we posit that autonomy will enhance the positive effects of performance goal clarity on performance. Employees who clearly understand the performance goals, when provided greater autonomy, will produce higher performance than others with lower autonomy.

**Hypothesis 3 (H3).** *Autonomy will positively moderate the relationship between performance goal clarity and performance.*

2.3.3. The Effects of Control Variables

We controlled for key work attitudes that are likely to affect performance—public service motivation and job satisfaction—and individual characteristics including occupational category (administrators or technicians), supervisory status, tenure, education, and demographic factors such as age and gender. Although not empirically consistent, scholars in public management argued that public employees with higher public service motivation are likely to perform better than others [4,46]. The relationship between job satisfaction and performance is also not clearly defined. In general, job satisfaction is predicted to be positively related to performance [4].

In the rank-in-person system, organizational tenure, which is likely to be significantly correlated with seniority, tend to be positively associated with performance evaluations. Individuals with longer seniority are more likely to receive higher performance ratings and also be eligible for promotion. The majority of the public workforce in the rank-in-person system is composed of general administrators, which may result in higher competitiveness among them than technicians. Human capital such as educational attainment and supervisory status can be positively related to performance. Performance of female employees, who are minorities in the organization, may be underestimated and possibly receive less favorable performance ratings than their male colleagues. Figure 1 describes the hypothesized model.

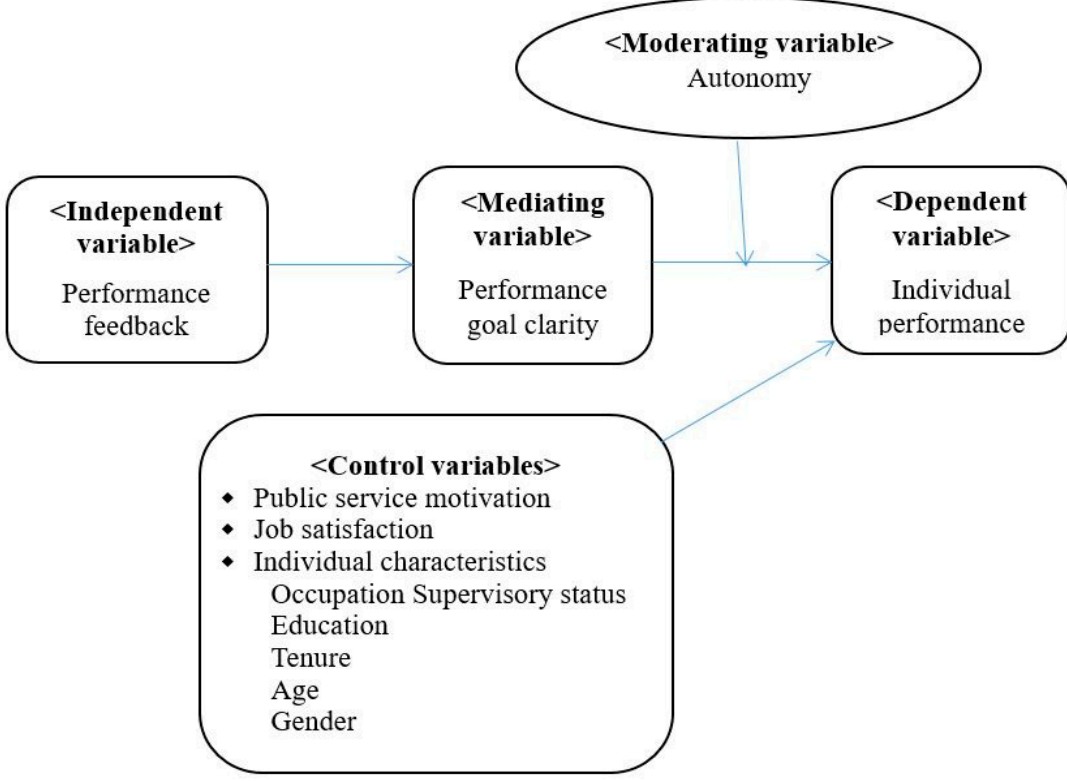

**Figure 1.** The mediated–moderated relationship between performance goal clarity and individual performance.

## 3. Data and Methods

### 3.1. Sample and Data Collection

The sample was drawn from the local government workforce in Korea. [1] The local government of Korea is composed of 17 metropolitan governments and 225 municipal governments. The disproportional

stratified random sampling method was used to select a sample, which was reliable and representative of the local government workforce in Korea. Considering different sizes of metropolitan and municipal governments, 30 units from each metropolitan government and 10 units from each municipal government were selected.

The anonymous survey was conducted 16 November 2017 to 5 February 2018. The computer-aided web interview (CAWI) method was adopted to collect the data from participants. We initially reached out individuals by telephone to ask if he or she is willing to participate in our survey before conducting it. Then, we emailed out our survey only to the ones who accepted our invitation; 2766 out of 8817 initial contacts completed the survey, leading to 31.4% of the response rate. [2] We also attempted to contact the ones who did not respond to our request several times to encourage them to participate in the survey. The survey items were developed to inquire of organizational productivity, individual and agency level performance ratings, performance evaluations and feedbacks, leadership, work attitudes, hierarchy and work autonomy, and demographic and personal characteristics of a survey respondent.

Table 1 displays the characteristics of the sample. The sample was composed of administrators (73.1%) and technicians (26.9%). [3] Approximately 60% of the survey respondents held rank 6 (30.5%) or rank 7 (29.4%); 10.4% of them are on rank 5, 18.5% on rank 8, 10.5% on rank 9, and 0.7% on rank 4. The average tenure of participants was 15.4 years. About 71% were college graduates. Women comprised 38.7% of the sample. Compared to the characteristics of the population, we found that our sample was representative of the population, sharing similar characteristics. The overall characteristics of the population were as follows: Men (58.6%) and women (41.4%); administrators (65%) and technicians (35%); rank 4 (1%), rank 5 (6.4%), rank 6 (28.1%), rank 7 (32%), rank 8 (15.5%), and rank 9 (11.5%); and average tenure (16.3 years).

**Table 1.** Descriptive statistics.

| Variable | Mean | Std. Dev | Min | Max | Unit of Analysis |
|---|---|---|---|---|---|
| Individual performance | 4.76 | 1.53 | 1 | 7 | Responses to the question, "what is your performance rating?" in a 7-point Likert scale: from 'outstanding = 7' to 'need improvement = 1' |
| Performance feedback | 2.56 | 1.40 | 1 | 5 | The number of times to meet and discuss with supervisor regarding performance last year (1 to 5 times) |
| Performance goal clarity | 3.34 | 0.79 | 1 | 5 | Index variable: the average score of responses to five survey items which were measured in 5-point scale |
| Autonomy | 3.03 | 0.70 | 1 | 5 | Index variable: the average score of responses to two survey items which were measured in 5-point scale |
| Public service motivation | 3.83 | 0.55 | 1 | 5 | Index variable: the average score of responses to seven survey items regarding satisfaction with job itself and public service motivation, each of which was measured in 5-point scale |
| Job satisfaction | 2.92 | 0.71 | 1 | 5 | Index variable: the average score of responses to seven survey items regarding satisfaction with salary, work hours, work environment, and employee welfare, each of which was measured in 5-point scale |
| Occupational category | 1.27 | 0.44 | 1 | 2 | General administrators = 1; technicians = 2 |
| Supervisory status | 1.11 | 0.31 | 1 | 2 | Non-supervisor = 1; manager = 2 |
| Educational attainment | 2.83 | 0.72 | 1 | 5 | PhD = 5; Master = 4; Bachelor = 3; 2 year college graduate = 2; High school graduates = 1 |
| Tenure | 4.20 | 3.52 | 0 | 11 | The number of years employed as public employees |
| Gender | 1.39 | 0.49 | 1 | 2 | Male = 1; female = 2 |

*3.2. Dependent Variable*

Individual-level performance. By law, the Korean government requires each agency to evaluate individual-level and organizational-level performance on a regular basis. [4] Individual performance is assessed by two different evaluation systems, depending on their rank. Performance of employees who hold rank 4 or higher is evaluated once per year (on the last day of a year), based on the performance agreement that they establish with their supervisor at the beginning of the year. Performance of employees who hold rank 5 or lower ranks is evaluated by their supervisor twice per year, in June and December (the last day of the year). The survey respondents were asked to report the performance ratings they received from the evaluator at the end of the year (12/31). The local governments in Korea adopt a 7-point rating scale to evaluate individual performance, which ranges from 'outstanding' to 'need improvement.' In the case of using different rating scales, they were asked to select one, which best describes the level of their performance evaluation. The average performance rating reported by the survey participants is 4.7, which indicates the performance level between 'moderately good' and 'average.' Although the measure still relies on self-reported data, the way of inquiring information can reduce the potential bias caused by self-assessment of performance. The survey participants were asked to report the performance rating they obtained from their supervisor, not their perceived level of performance.

*3.3. Independent Variables*

3.3.1. Performance Feedback

The variable was developed based on responses to the question, "how many times did you discuss your performance with your supervisor during this year? [5] We assume that the more frequently an individual received performance feedback from his or her supervisor, the more information regarding performance goals, which he or she should attain, and the expected levels of performance, would have been provided. The average number of performance feedback provided for employees is 3 times this year, ranging between 1 and 5 times.

3.3.2. Performance Goal Clarity

The perceived level of performance goal clarity was measured by combining responses to relevant questions. Sawyer (1992) developed the measures of goal clarity, which inquire of clear understanding of duties and responsibilities, goals and objectives for the job, the relationship between individual work and the overall objectives of the work unit, the expected results of my work, and information of my work to get positive evaluations (or avoid negative evaluations) [47]. Referring to Sawyer's (1992) measures of goal clarity, we selected relevant survey items that specifically focus on measuring performance goals clarity. Five items inquiring of the following subjects were included: (1) Objectivity and measurability of performance goals; (2) a clear understanding of performance goals; (3) relationship between individuals' performance goals and the organization's goals; (4) a clear understanding of goal priorities; (5) information to avoid poor performance ratings. All these items were measured on a 5-point Likert scale. Cronbach's alpha value (0.90) suggests that the measure is internally consistent and reliable. We calculated the average scores of the responses to five survey items. (see Appendix A)

3.3.3. Autonomy

This variable measured the perceived level of autonomy employees exercise in their work. Campbell and Pritchard (1976) developed the original measure of work autonomy [48]. Because many of the original items specifically focus on managerial roles, we selected two appropriate items for our sample, 89% of which were non-supervisors. The measures evaluated the extent to which employees felt free to determine their work processes, schedule tasks, and any work-related decisions. The questions that were asked concerned whether employees have higher levels of autonomy in their work and if a supervisor frequently delegates authority to subordinates. They were measured on a

5-point Likert scale. The responses to these questions (two survey items) were correlated, leading to a Cronbach's alpha value of 0.65. It suggests that the variable is internally consistent and reliable. The responses to the questions were combined by calculating the average scores of the responses to the survey items.

### 3.4. Control Variables

#### 3.4.1. Work Attitudes

We controlled for the influences of public service motivation and job satisfaction on performance. The seven survey items of public service motivation were developed based on Perry's (1996) public service motivation measures. The measure of job satisfaction was composed of seven survey items including satisfaction with pay, working conditions, environment, and welfare programs. Each variable was measured by averaging responses to the relevant questions. Cronbach's alpha for the measure of job satisfaction was 0.73, while it was 0.85 for the measure of public service motivation; it shows that the measures are internally consistent.

#### 3.4.2. Individual Characteristics

To take into consideration the impacts of individual differences on performance, we controlled for six variables measuring individual characteristics: Occupation, tenure, educational attainment, gender, and supervisory status. The occupation variable had two values: administrator coded as "1," and technician coded as "2." Regarding this, 73% of the sample was administrators, while the rest of the sample was technicians. The tenure variable measured the number of years an individual has worked in government. For the gender variable, men were coded as "1," while women coded as "2." The supervisory status variable had two values: managers coded as "2" and non-supervisor coded as "1." The educational attainment variable had five values: PhD degree coded as "5," Master degree coded as "4," Bachelor degree as "3," 2-year college graduates coded as "2," and high school graduate coded as "1." Table 1 displays the descriptive statistics, while Table 2 reports bivariate correlations among variables.

**Table 2.** Correlations.

| | 1 | 2 | 3 | 4 | 5 | 6 | 7 | 8 | 9 | 10 |
|---|---|---|---|---|---|---|---|---|---|---|
| 1. Occupation | | | | | | | | | | |
| 2. Supervisory status | −0.02 | | | | | | | | | |
| 3. Tenure | −0.01 | 0.04 * | | | | | | | | |
| 4. Education | 0.02 | 0.02 | 0.04 * | | | | | | | |
| 5. Gender | −0.21 ** | −0.19 ** | −0.00 | 0.1 ** | | | | | | |
| 6. Feedback | 0.02 | 0.14 ** | 0.02 | 0.03 | −0.16 ** | | | | | |
| 7. Performance goal clarity | 0.10 ** | 0.12 ** | 0.02 | −0.01 | −0.07 ** | 0.37 ** | (0.71) | | | |
| 8. Autonomy | 0.05 * | 0.08 ** | 0.01 | 0.01 | −0.06 ** | 0.19 ** | 0.30 ** | (0.71) | | |
| 9. Public service motivation | 0.03 | 0.17 ** | 0.09 ** | −0.01 | −0.11 ** | 0.21 ** | 0.36 ** | 0.21 ** | (0.57) | |
| 10. Job satisfaction | −0.01 | 0.20 ** | 0.04 * | −0.05 ** | −0.04 ** | 0.21 ** | 0.36 ** | 0.27 ** | 0.27 ** | (0.5) |

Note: The average variance extracted (AVE) values were in the parentheses. **, $p < 0.01$; *, $p < 0.05$.

### 3.5. Model Specification and Testing Methods

We developed the models that propose the effect of performance feedback on individual performance mediated by performance goal clarity and moderated by work autonomy. Although the data were collected at one point in time, the way we structured the questions may have created a natural time lag between performance feedback and performance evaluation. The survey participants were likely to report their performance rating that they received at the end of the year, given the survey was conducted during the performance evaluation period. Then, we can reasonably expect that performance feedbacks, which individuals received during a year, influenced the evaluation results obtained on the last day of the year.

Before testing the hypotheses, the confirmatory factor analysis was conducted to evaluate the measurement model for convergent and discriminant validity. The convergent and discriminant validity of the focal constructs in our models, which include performance goal clarity, autonomy, job satisfaction, and public service motivation, were examined through a series of confirmatory factor analyses. Standardized loading estimates for all the items in the four constructs ranged between 0.52 and 0.93. The average variance extracted (AVE) values for all four variables (in Table 2) were over 0.5, which is the recommended cutoff point [49,50]. The AVE of performance goal clarity corresponds to 0.71; autonomy, 0.71; job satisfaction, 0.5; public service motivation, 0.57. Composite reliabilities of four variables were 0.91 (performance goal clarity), 0.83 (work autonomy), and 0.87 (public service motivation). The composite reliability of job satisfaction was 0.67, which was lower than the recommended cutoff point (0.70), but an acceptable level [51].

To test discriminant validity, we tested four models: one-factor model to four-factor and compared the fit indices of the hypothesized models. The results are shown in Table 3. In the one-factor model, all the variables were loaded on a single factor. In the two-factor model, autonomy, job satisfaction, and public service motivation were loaded on one factor. In the three-factor model, job satisfaction and public service motivation were loaded on one factor. In the four-factor model, each variable was loaded on a single factor. The hypothesized four-factor model was shown to provide a better fit than the other models. The fit indexes including 683.43 (chi-square), 0.96 (CFI), 0.03 (SRMR), and 0.06 (RMSEA) are indicative of acceptable fit [52,53]. Table 2 displays that the AVE of each construct was greater than its shared variance with any other construct, suggesting that discriminant validity was supported for the four constructs [50].

**Table 3.** The comparison of the measurement models.

| Models | $\chi^2$ | df | RMSEA | CFI | SRMR |
|---|---|---|---|---|---|
| 4-factor model | 683.43 *** | 59 | 0.062 | 0.962 | 0.03 |
| 3-factor model | 985.8 *** | 62 | 0.073 | 0.943 | 0.035 |
| 2-factor model | 1634.83 *** | 64 | 0.094 | 0.904 | 0.057 |
| 1-factor model | 6714.3 *** | 65 | 0.192 | 0.593 | 0.132 |

Note: 1-factor model (performance goal clarity, autonomy, job satisfaction, and public service motivation (PSM) → one factor); 2-factor model (performance goal clarity and autonomy→ one factor, PSM and job satisfaction → the other factor); 3-factor model (PSM and job satisfaction→ one factor, performance goal clarity → another factor, autonomy → the other factor); 4-factor model, each variable was loaded on a single factor. ***, $p < 0.001$.

To test the hypothesized relationships, structural equation modeling (SEM) was performed. To examine the indirect effects of performance feedback on individual performance through performance goal clarity, we adopted the bootstrap estimation method, using 1000 replications. Asymmetric bootstrap confidence intervals have been widely used to test indirect effects [54]. Evidence of 95% bootstrap confidence intervals that are above zero indicates the statistical significance of indirect effects [54]. We also included a multiplicative term of performance goal clarity and autonomy in the models, to test the moderating effect of autonomy on the relationship between performance goal clarity and performance. We mean-centered each constituent variable before generating the multiplicative term (or interaction variable). To probe the moderation effect, we conducted a simple slope analysis by testing the effect of goal clarity on individual performance at the low level of autonomy (one standard deviation (SD) below the mean) and at the high level of autonomy (one SD above the mean) [55,56]. The result was plotted.

## 4. Results

Figure 2 provides the summary of the path estimates between feedback, goal clarity, autonomy, and individual performance, where the effects of work attitudes (job satisfaction and public service motivation) and individual characteristics were controlled. The fit indices of the model show that the model fit was good: The Chi-square = 4.1 ($p < 0.1$), RMSEA = 0.034, CFI=0.998, SRMR = 0.004 [54].

Table 4 shows the detailed estimates of structural path estimates. When testing the hypothesized model, we compared it with alternative models to determine a direct relationship between performance feedback and individual performance. The alternative model 1 deleted the direct effect of performance feedback and individual performance. The fit indices of the alternative model included the Chi-square = 365.9 ($p < 0.001$), RMSEA = 0.166, CFI = 0.74, SRMR = 0.043 (Table 5). The alternative model 2 added the indirect paths of autonomy, generic work motivation, and public service motivation on individual performance. The fit indices of the alternative model included the Chi-square = 18.44 ($p < 0.001$), RMSEA = 0.056, CFI = 0.98, SRMR = 0.007 (Table 5). We, thus, verified the validity of our hypothesized model. Table 5 displays the summary of the fit indices of the hypothesized model and alternative models.

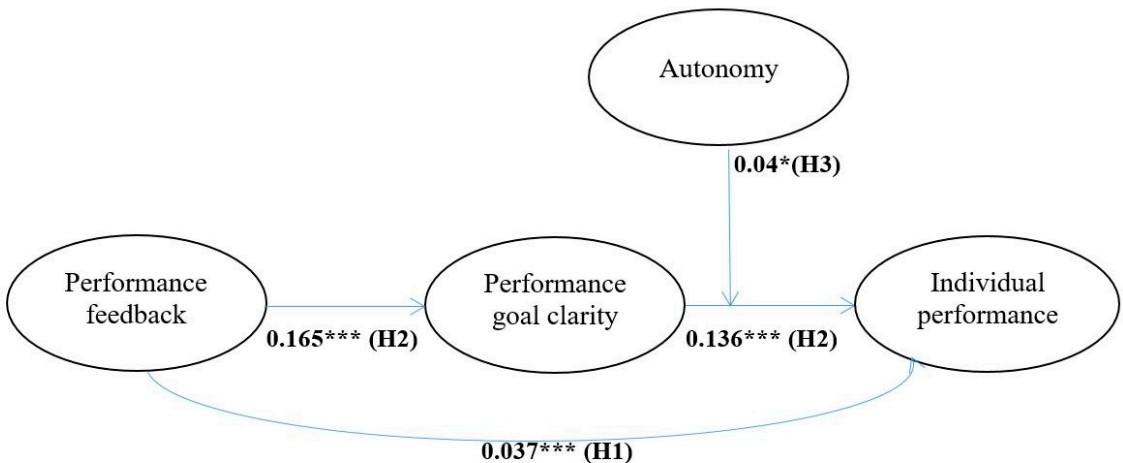

**Figure 2.** The results of the mediated–moderated relationship between performance goal clarity and individual performance. ***, $p < 0.001$; *, $p < 0.1$.

**Table 4.** The estimated structural path coefficients (N = 2630).

| Structural Path | Coef. | Std. Err. |
|---|---|---|
| Performance feedback → performance goal clarity | 0.165 *** | 0.01 |
| Autonomy → performance goal clarity | 0.213 *** | 0.02 |
| Job satisfaction → performance goal clarity | 0.098 *** | 0.03 |
| Public service motivation → performance goal clarity | 0.277 *** | 0.03 |
| Performance goal clarity → individual performance | 0.136 *** | 0.02 |
| Performance feedback → individual performance | 0.037 *** | 0.01 |
| Autonomy → individual performance | 0.069 *** | 0.02 |
| Job satisfaction → individual performance | 0.178 *** | 0.03 |
| Public service motivation → individual performance | 0.159 *** | 0.03 |
| Performance goal clarity*autonomy → individual performance | 0.041 * | 0.02 |
| Occupation → performance goal clarity | 0.146 *** | 0.03 |
| Tenure → performance goal clarity | 0.0003 | 0.002 |
| Education → performance goal clarity | −0.025 | 0.02 |
| Gender → performance goal clarity | 0.097 ** | 0.03 |
| Supervisory status → performance goal clarity | 0.072 | 0.05 |
| Occupation → individual performance | 0.014 | 0.03 |
| Tenure → individual performance | 0.006 *** | 0.001 |
| Education → individual performance | 0.029 | 0.017 |
| Gender → individual performance | −0.08 ** | 0.03 |
| Supervisory status → individual performance | −0.018 | 0.05 |

Note: $\chi^2$ (1) = 4.1 *, RMSEA = 0.034, CFI = 0.998, SRMR = 0.004, ***, $p < 0.001$; **, $p < 0.01$; *, $p < 0.1$.

**Table 5.** The summary of fit indices of the hypothesized model and alternative models.

| Models | $\chi^2$ | df | RMSEA | CFI | SRMR |
|---|---|---|---|---|---|
| The hypothesized model | 4.1 * | 1 | 0.034 | 0.998 | 0.004 |
| Alternative model 2 | 18.44 *** | 2 | 0.056 | 0.988 | 0.007 |
| Alternative model 1 | 365.9 *** | 5 | 0.166 | 0.74 | 0.043 |

Note: ***, $p < 0.001$; *, $p < 0.1$.

H1 posits that performance feedback will be positively related to individual performance. The direct relationship between performance feedback and individual performance was found to be positive (0.037, $p < 0.001$), which is consistent with our expectations. It suggests that more frequent feedback on performance was positively associated with the enhanced performance of employees. Individuals who had more opportunities to meet their supervisor and communicate performance goals, performance information, and expected behaviors were more likely to attain higher evaluation ratings of performance.

H2 postulates that performance goal clarity would mediate the relationship between performance feedback and performance. We expected that more frequent feedback would help employees better understand the performance goals and expectations they should meet, in turn improving performance. Especially in the public sector where multiple goals compete and conflict with each other, performance feedback will play a critical role in prioritizing the goals and guide employees in the way that they can select goals and concentrate their efforts on the organizational priorities. To test the mediation effect, we performed the bootstrapping estimation. The results in Table 6 show that the indirect effect of performance feedback on individual performance through performance goal clarity was also significant (0.022, $p < 0.001$, 95% confidence interval [0.015, 0.030]), which is supportive of H2. Performance feedback was positively related to higher clarity of performance goals (0.165, $p < 0.001$). Higher goal clarity was also positively related to individual performance (0.136, $p < 0.001$). They indicate that performance goal clarity significantly mediates the relationship between performance feedback and individual performance. More frequent feedback and discussion on performance will first clarify performance goals for employees, in turn aiding in their higher performance. The indirect relationships between autonomy, job satisfaction, public service motivation, and individual performance, which are mediated by performance goal clarity, were also significant. Table 6 displays the results of the bootstrap estimation.

**Table 6.** The results of the bootstrap estimation.

| Indirect Effect | Coef. | Bootstrap Std. Err. | 95% Confidence Interval |
|---|---|---|---|
| Performance feedback → performance goal clarity→ individual performance | 0.0224 *** | 0.004 | [0.015, 0.030] |
| Autonomy→ performance goal clarity→ individual performance | 0.029 *** | 0.005 | [0.019, 0.039] |
| Job satisfaction→ performance goal clarity→ individual performance | 0.013 ** | 0.004 | [0.005, 0.022] |
| Public service motivation→ performance goal clarity→ individual performance | 0.038 *** | 0.007 | [0.024, 0.052] |

Note: 1000 replications, ***, $p < 0.001$; **, $p < 0.01$.

H3 assumes that autonomy will positively moderate the relationship between performance goal clarity and performance in the way that employees, who clearly understand the performance goals, when provided a higher level of work autonomy, will produce higher performance than others with lower autonomy. The moderation effect of autonomy on the relationship between performance goal clarity and individual performance was significant (0.041, $p < 0.05$). The hypothesis was supported, suggesting that employees with higher autonomy will benefit more from performance goal clarity,

achieving higher performance. Figure 3 plots the moderation effect of autonomy on the relationship between goal clarity and individual performance.

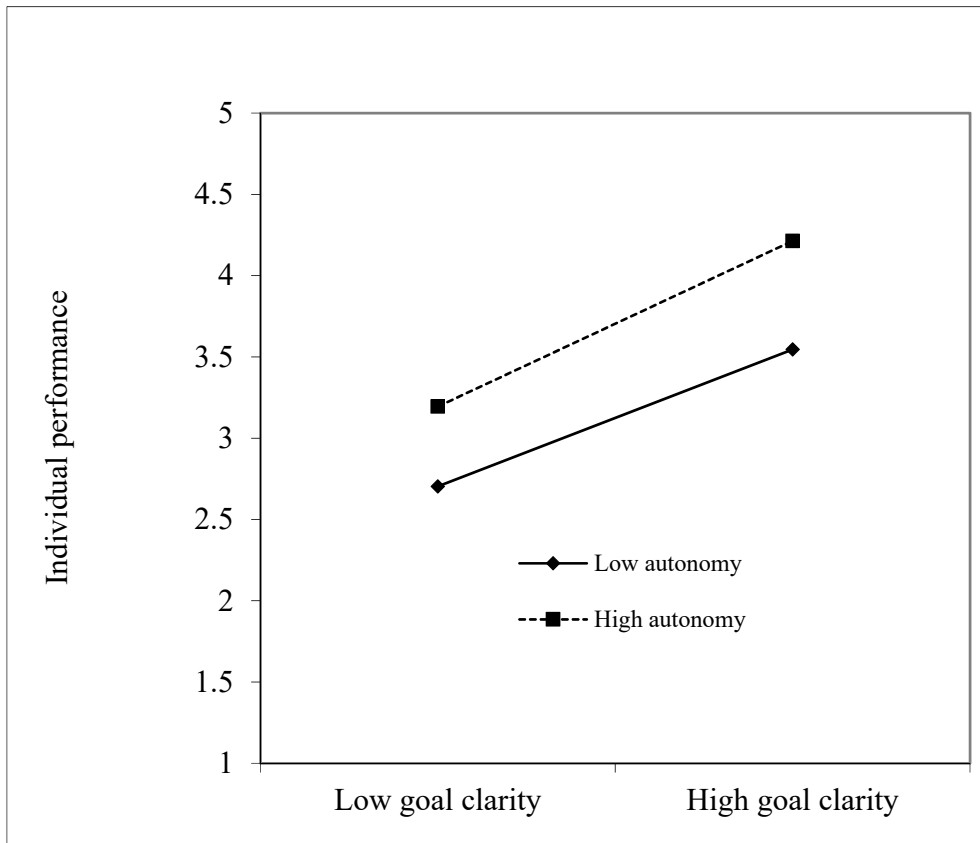

**Figure 3.** Plotting the moderation effect of autonomy.

The models control the effects of work attitudes and individual characteristics on individual performance. Job satisfaction (0.178, $p < 0.001$) and public service motivation (0.159, $p < 0.001$) were both positively associated with individual performance. In addition, the relationships between job satisfaction and public service motivation, and performance were mediated by goal clarity like that between feedback and performance. Employees with longer tenures were more likely to attain higher performance ratings. It might be attributed to the characteristics of the personnel system in the Korean government, which has heavily relied upon seniority in making important personnel decisions including assignment of works and roles, promotion, and wage. [8] Tenure, thus, is inclined to have a positive effect on a performance rating. Female employees tended to receive lower performance ratings. The negative correlation between tenure and women may indicate that female employees have shorter tenures than their male colleagues, which may lead to relatively lower performance ratings for women. Occupational category, supervisory status, and educational attainment were not significantly related to individual performance.

## 5. Discussion and Conclusions

Prior research has investigated how performance feedback can contribute to individual performance [14]. Empirical evidence has not consistently indicated that feedback on performance positively influences individuals' job performance [14,24,27,28]. Such inconsistency may suggest the existence of a dynamic relationship between feedback and performance, which might be influenced by a variety of contextual factors. Indeed, our research has shown that performance feedback may contribute to the improvement of individual performance by clarifying the performance goals they

need to focus on. We also found that when an individual has been given higher autonomy in their work, while also having a clearer sense of the organizational and performance goals, he or she could further improve their performance. These findings support theoretical arguments of the potential benefits of feedback over performance improvement.

This research adds to the literature in some meaningful ways. First, little research has analyzed the potential dynamic relationships between feedback, performance goal clarity, autonomy, and individual performance in the context of public organizations. This study has demonstrated that promoting a feedback-rich environment and supporting a coaching approach to performance management can possibly alleviate the chronic problem of goal ambiguity in the public sector, ultimately enhancing public employees' productivity, and organizational performance. Furthermore, it highlighted the importance of employees' work autonomy, which can boost the positive effects of performance feedback and performance goal clarity on individual performance. Given public employees' autonomy in their work has often been considerably limited by the complicated sets of legal and political constraints, our findings will have an important practical implication on effective performance management in the public sector. Second, an objective measure of individual performance was adopted to test the feedback effects. Some previous studies used perceptual measures of feedback and performance or attitudinal outcomes to test the feedback effects, which might cause the results to be highly vulnerable to mono-source bias [14]. We have improved the robustness of the results by using a more objective way of measuring performance feedback and individual performance. Performance evaluation results and the actual number of feedback experiences were adopted to test the relationships.

The primary finding of this study is that feedback can help public employees overcome the challenges of goal ambiguity, ultimately attaining higher performance evaluations. The result is consistent with the arguments of goal-setting theory [23] and control theory, which noted that in the process of accomplishing goals, feedback plays a guiding role in directing individual workers to follow the behavioral standards and expectations valued by the organization and to pay attention to the aspects of tasks indicated by feedback. Our findings showed that employees who had more chances to receive feedback from their supervisor were likely to accomplish higher performance evaluations than others. It suggests that as theoretical arguments indicated, feedback on performance will improve individual performance by providing proper instructions and guidelines for employees to obtain desirable outcomes for both employees and the organization. As a result, performance feedback can help individuals set future goals and behaviors in the direction of promoting their productivity, therefore contributing to higher performance of the organization [25].

Another interesting finding is the moderation effect of autonomy, which may influence the relationship between performance goal clarity and individual performance. According to the result, employees who have a higher level of autonomy on their work are likely to take greater advantage of a clear perception of organizational expectations over individual performance and behaviors. Autonomy allows individuals to retain control over how to channel their efforts towards high performance and to accomplish goals congruent to the organization's [11]. Employees' voluntary efforts towards goal accomplishment, armed with a firmer insight into what goals the organization want employees to accomplish and how they behave, will generate synergic effects on organizational goal attainment [11,14,44]. In a similar vein, Locke and Latham (1990) in goal-setting theory also argued that autonomous employees can select goals consistent with those of the organization and invest more effort in meeting with the organization's expectations over them [23].

The results of this study provide important practical implications for employee development and performance management in the public sector. Performance evaluations, which are rarely conducted only once or twice per year, will not be sufficient for helping employees in improving their performance [14]. Supervisory feedback, either formally or informally, can fill in the gaps between employee demand on feedback and formal performance evaluations and feedbacks [13]. In addition, more opportunities for feedback may help employees to obtain development-related advice on a more consistent basis [14,57], which will ultimately contribute to their career development as well as

organizational effectiveness. Encouraging employees to seek feedback on their work and behaviors will create a feedback environment where the overall level of employees' understanding of organizational goals and behavioral expectations is elevated. Eventually, such organizational culture will facilitate the organization's performance management practices and developing its highly performing workforces.

Further research is required to address the limitations of this study. We tested our hypotheses by analyzing cross-sectional data, which may be limited in establishing the causal link between performance feedback and individual performance. Although not relying completely on respondents' perceptions, our measure of individual performance was developed based on self-reported data. Self-reported performance ratings may be less accurate than those provided by evaluators; said otherwise, the potential gaps may exist between self-reported ratings and actual performance ratings. Nevertheless, it still reduces the potential mono-source bias, which may be associated with self-assessment of performance. Future research could improve such limitations by developing longitudinal research designs with more objective measures of individual performance.

**Author Contributions:** Data curation, S.P.; Methodology, S.C.; Writing-Original Draft, S.P. and S.C. All authors have read and agreed to the published version of the manuscript.

**Funding:** This study was financially supported by the Public Performance Management Research Center in the Graduate School of Public Administration at Seoul National University. [Project Number: 0678-20180008].

**Conflicts of Interest:** The authors declare no conflict of interest

### Appendix A

◆ Performance feedback

  - How many times did you discuss about your performance with your supervisor last year?

◆ Performance goal clarity (alpha = 0.88; 5 point scale from "very disagree to very agree")

  - Individuals clearly understand their performance goals.
  - Individuals' performance goals can be objectively measured.
  - Individuals' performance goals are clearly ordered by their priority.
  - Individuals' performance goals are properly aligned with organizational goals.
  - The reasons for an individual's poor performance evaluation are clearly explained.

◆ Autonomy (alpha = 0.65; 5 point scale from "very disagree to very agree")

  - Individuals have a high level of work autonomy.
  - Supervisors often delegate work authority to their subordinates.

◆ Public service motivation (alpha = 0.87; 5 point scale from "very disagree to very agree")

  - I am strongly committed to work.
  - I do my best with very challenging works.
  - I prioritize the interest of the local community over my private interest.
  - I feel strong accountability for the society.
  - I can sacrifice myself to help others.
  - I feel sympathetic for people in difficult situation.
  - I feel good when my idea contribute to public policy.

◆ Job satisfaction (alpha = 0.87; 5 point scale from "very dissatisfied to very satisfied")

  - I am satisfied with salary.
  - I am satisfied with workload.
  - I am satisfied with work hours.
  - I am satisfied with performance pay.

- I am satisfied with employee welfare.
- I am satisfied with work environment.
- I am satisfied with training and education.

**Notes:**

1. The Korean government has the rank-in-person system that requires specialized skills and expertise relatively less than other personnel systems [58]. Public employees often rotate different jobs and learn different skills. They traditionally acquire necessary skills and knowledge that are required to perform their duties from their supervisors and previous job holders. Performance appraisals also are conducted weighing much on evaluators' subjective assessments. Thus, employees are more likely to rely on performance feedbacks from their supervisor that are perceived to be directly connected to performance appraisals than any job-related documentation (e.g., job descriptions, manuals).
2. When disaggregated to metropolitan and municipal governments, the response rates correspond to 18.7% (509 out of 2722) and 37% (2257 out of 6095), respectively.
3. The rank-in-person system offers the basic personnel system in the Korean government. The Korean public personnel system preferably hires generalists who have general administrative ability and knowledge rather than specialists who have special skills and expertise. There are three types of occupations in the Korean government: (1) General administrators, (2) specialists (e.g., police, teachers), and (3) political appointees and supporting positions. The general administrator category is comprised of administrators (65%) and technicians (35%). The population of the survey corresponds to the group of general administrators in the local governments in Korea.
4. Individual performance is evaluated twice per year in June and December.
5. Because the survey was conducted around the end of the year (November 2017–February 2018), the question asks how many times a respondent obtained performance feedbacks during the year of 2017.
6. Comparative fit index (CFI) values range from 0 to 1. A CFI value of 0.9 or larger indicates acceptable model fit. The standardized root mean square residual (SRMR) ranges from 0 to 1, with a value of 0.08 or less indicating an acceptable model. The root mean square error of approximation (RMSEA) ranges from 0 to 1 and smaller values of RMSEA indicate better model fit. A value of 0.06 or less indicates acceptable model fit [53].
7. The values of covariance between the four construct are 0.25 (performance goal clarity and work autonomy), 0.15 (performance goal clarity and job satisfaction), 0.18 (performance goal clarity and public service motivation), 0.09 (work autonomy and job satisfaction), 0.1 (work autonomy and public service motivation), and 0.23 (job satisfaction and public service motivation).
8. Tenure can serve as an important determinant of employee promotion. Employees are inclined to have higher performance ratings when their turn for promotion comes.

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
