# Peer review of "Performance Feedback, Goal Clarity, and Public Employees’ Performance in Public Organizations"

_sustainability, doi:10.3390/su12073011_

Round 1

Reviewer 1 Report

Dear authors, 

In this study relationship between performance feedback on performance and individual performance is examined. There are similar studies, but the uniqueness of this study is reflected in the fact that it is intended for public employees. Paper is logically structured and all chapters have corresponding order. Sections Theoretical framework, as well as Data and methods, are providing enough information regarding design of the study, sampling etc. Hypotheses are clearly described as well are theoretically well-grounded. Results are clearly presented and the conclusion section is supported by the content. 

In my opinion, there are two shortcomings that need to be corrected in order for the paper to be acceptable:

-In the introduction, you need to mention maybe some ways and methods of giving feedback. There are many ways as well as methods, and sometimes this is crucial for the employee i.e. the way he is receiving FB and for the superior the way he is giving FB. Please back this up!

-My second and I must say the major concern is regarding the relevance of your literature review. I must say that most of the references are outdated. Indeed if it is needed, you should mention some older ref., but you didn't have any reference from 2019, and you have only one from 2018, none from 2017...  This part needs to be significantly improved!

Author Response

Memorandum of Revision

Thank you very much for your careful review of our manuscript. The revisions you suggested required a thorough effort on our part. We believe that the manuscript has substantially improved paying attention to your suggestions. If your concerns still remain, please feel free to ask further revision. I look forward to your reactions and comments to the revision. The detailed changes we made in our manuscript are as follows.

<Responses to Reviewer 1>

Comment 1

  1. In the introduction, you need to mention maybe some ways and methods of giving feedback. There are many ways as well as methods, and sometimes this is crucial for the employee i.e. the way he is receiving FB and for the superior the way he is giving FB. Please back this up!

Response to comment 1

We discussed the different ways of offering feedbacks and relevant empirical findings of the different impacts on job performance in the theoretical framework section. Please review the 7th line in the first paragraph on p. 4—the 10th line from the top of page 5.

Comment 2

  1. My second and I must say the major concern is regarding the relevance of your literature review. I must say that most of the references are outdated.

Response to comment 2

We thoroughly revised our literature review adding up more recent research (e.g., Aljadeff-Abergel et al. 2017; Choi et al. 2018; Favero et al. 2016; Guo et al. 2014; Johnson, Rocheleau, and Tilka 2015; Nicholson-Crotty, Nicholson-Crotty, and Fernandez 2017; Su et al. 2019; Zheng et al. 2013; Zhou 2003). Please review the revised theoretical framework and hypotheses section (the underlined on p. 4—7).

Reviewer 2 Report

Dear Editor,

Thanks for the opportunity to review the manuscript titled: “Performance feedback, goal clarity, and public  employees’ performance: Testing the impact of  supervisory feedback on individual performance".

The manuscript addresses an issue of supervisory feedback impact on employee performance. Authors examine the mediating effect of performance goal clarity in this relationship. The topic itself is interesting. However, the manuscript need minor revisions to be consider for publication.

Therefore, I would suggest the following recommendations, which will greatly improve the manuscript:

General points
1. From my point of view the bibliographical references are out of date. The majority of (36 out of 46) bibliographic references (78%)  are from the last decade. Furthermore, more than a half are form the past century (26 out of 46). Only 9 out of 46 is form recent decade (19%).  It is necessary to update and improve the bibliographical references. Additionally the used literature should be much more extended. The references in this paper don’t provide sufficient background.

  1. I would recommend for authors to significantly extend theoretical framework.

  1. The topic is interesting, however not novel.

In the introductions section authors should explain , why this approach is unique and how this contribution differs from others. The justification should be clearly provided. ( there are many current studies analyzing this subject, for instance : https://doi.org/10.1016/j.hrmr.2019.100740 ; https://doi.org/10.1080/14330237.2019.1665879, https://doi.org/10.1108/LODJ-04-2013-0039 ; https://doi.org/10.2224/sbp.2014.42.5.731)

I suggest authors will suggest how their research distinguish. And broadly place their research  in an up-to date literature.

  1. I would recommend describing the organization of a paper in the introduction

  1. Sampling: The  authors should mention whether or not an incentive was offered for completion of the questionnaire. Were study participants assured for confidentiality?

  1. Methods are not sufficiently described. Authors should refer to the current literature and guidance.

  1. Mediation analysis: I would suggest to indicate confidence intervals for indirect effects in mediation analysis (Hayes, A. F. (2012). PROCESS: A versatile computational tool for observed variable mediation, moderation, and conditional process modelling )

  1. Moderation analysis: I would suggest adding slopes that illustrate the effect

  1. The limitation of this paper should be presented. One paragraph should depicts possible shortcomings.

  1. In my opinion Authors should provide contribution of this study. This is an important element of each paper.

In my opinion, the manuscript  needs major revision. In conclusion, this paper is good, but some improvements are necessary for the final publication.

Author Response

Memorandum of Revision

Thank you very much for your careful review of our manuscript. The revisions you suggested required a thorough effort on our part. We believe that the manuscript has substantially improved paying attention to your suggestions. If your concerns still remain, please feel free to ask further revision. I look forward to your reactions and comments to the revision. The detailed changes we made in our manuscript are as follows.

<Responses to Reviewer 2>

Comment 1

It is necessary to update and improve the bibliographical references. Additionally the used literature should be much more extended. The references in this paper don’t provide sufficient background.

Response to comment 1

We thoroughly revised our literature review adding up more recent research (e.g., Aljadeff-Abergel et al. 2017; Choi et al. 2018; Favero et al. 2016; Guo et al. 2014; Johnson, Rocheleau, and Tilka 2015; Nicholson-Crotty, Nicholson-Crotty, and Fernandez 2017; Su et al. 2019; Zheng et al. 2013; Zhou 2003). Please review the revised theoretical framework and hypotheses section (the underlined on p. 4—7).

Comment 2

I would recommend for authors to significantly extend theoretical framework.

Response to comment 2

We extended our theoretical framework section by adding more discussion over performance feedbacks such as different ways of offering feedbacks and various functions of feedbacks along with the review of relevant research. In the hypotheses section, we focused more on the relationship between feedbacks and performance and added more recent research on the relationship (e.g., Aljadeff-Abergel et al. 2017; Choi et al. 2018; Favero et al. 2016; Guo et al. 2014; Johnson, Rocheleau, and Tilka 2015; Nicholson-Crotty, Nicholson-Crotty, and Fernandez 2017; Su et al. 2019; Zheng et al. 2013; Zhou 2003). Please review the revised theoretical framework and hypotheses section (the underlined on p. 4—7).

Comment 3

The topic is interesting, however not novel. In the introductions section authors should explain why this approach is unique and how this contribution differs from others. The justification should be clearly provided (https://doi.org/10.1016/j.hrmr.2019.100740 ; https://doi.org/10.1080/14330237.2019.1665879, https://doi.org/10.1108/LODJ-04-2013-0039 ; https://doi.org/10.2224/sbp.2014.42.5.731

Response to comment 3

We agree with the reviewer’s argument. But we believe that in public management there is less research on the impacts of performance feedbacks on individual performance. The literature the reviewer suggested above analyzed the data from private businesses. We cited them to update our literature review (Choi et al. 2018; Guo et al. 2014; Su et al. 2019; Zheng et al. 2013). In this research we aim to test if the findings from private management are also applicable to public organizations that are characterized by unclear organizational goals and political environments. To respond to the reviewer, we completely rewrote the introduction (p. 2—4). Throughout the introduction section, we made an argument of the necessity of research with the data from public workforces due to the uniqueness of the public sector. Please review the introduction. In the conclusion we also reemphasized the contribution of the research (the 2nd paragraph on p. 23—24).  

Comment 4

I would recommend describing the organization of a paper in the introduction.

Response to comment 4

Please review the 1st paragraph on p. 4.

Comment 5   

Sampling: The authors should mention whether or not an incentive was offered for completion of the questionnaire. Were study participants assured for confidentiality?

Response to comment 5
à Incentives of completing a survey are not allowed for public employees based on the anti-corruption law in Korea. The survey was anonymously conducted (the 1st line in the 3rd paragraph on p. 13).

Comment 6

Methods are not sufficiently described. Authors should refer to the current literature and guidance.

Response to comment 6

à Hayes (2012) and Dawson (2014) articles were added to extend the discussion of methods. Please review the revised methods section (the 2nd paragraph on p. 19—20).  

Comment 7 

Mediation analysis: I would suggest to indicate confidence intervals for indirect effects in mediation analysis (Hayes, A. F. (2012). PROCESS: A versatile computational tool for observed variable mediation, moderation, and conditional process modelling )

Response to comment 7

The discussion of the bootstrap results and CIs was added in the methods and results section (the 4th—6th line in the 2nd paragraph on p. 19; the 7th—10th line in the 2nd paragraph on p. 21). Please review Table 6 on p. 40.    

Comment 8

Moderation analysis: I would suggest adding slopes that illustrate the effect

Response to comment 8

The slopes were plotted. Please review Figure 3 on p. 42.

Comment 9

The limitation of this paper should be presented. One paragraph should depicts possible shortcomings.

Response to comment 9 

Please review the last paragraph on p. 26.

Comment 10

In my opinion Authors should provide contribution of this study. This is an important element of each paper.

Response to comment 10

Throughout the introduction section, we made an argument of the necessity of research with the data from public workforces due to the uniqueness of the public sector ((p. 2—4). Please review the introduction. In the conclusion, we also reemphasized the contribution of the research (the 2nd paragraph on p. 23—24).  

Reviewer 3 Report

  1. The study has a work psychological focus and in its present form has only an indirect link with the sustainability. The relevance of the sustainability should be emphasized both in the abstract, in the introduction and throughout the article.
  2. It is not right for authors to describe in the introduction what relationships are likely between variables without logically deducing its theoretical background. In the introduction, it is only necessary to highlight thoroughly why it is need to examine the results reported in this article.
  3. H1 is surplus. Many studies had already described this, see e.g. Kluger and DeNisi (1996) or Zhou (1998) in the Journal of Applied Psychology. H2 and H3 make two statements at the same time, which is methodologically incorrect.
  4. As the sample comes from the Korean government, it is necessary to write in detail why these relationships should be investigated in the public sector. What are the characteristics of the Korean organizational cultures in the government? How much feedback is there in relation to the competitive sector or internationally? How are performance expectations handled in government and companies?
  5. The method and the conclusions are acceptable.
  6.  

Author Response

Memorandum of Revision

Thank you very much for your careful review of our manuscript. The revisions you suggested required a thorough effort on our part. We believe that the manuscript has substantially improved paying attention to your suggestions. If your concerns still remain, please feel free to ask further revision. I look forward to your reactions and comments to the revision. The detailed changes we made in our manuscript are as follows.

<Responses to Reviewer 3>

 Comment 1

The study has a work psychological focus and in its present form has only an indirect link with the sustainability. The relevance of the sustainability should be emphasized both in the abstract, in the introduction and throughout the article.

Response to comment 1 

We argued that performance management is directly related to organizational sustainability. Please review the 1st paragraph in the introduction on p. 2.

Comment 2

It is not right for authors to describe in the introduction what relationships are likely between variables without logically deducing its theoretical background. In the introduction, it is only necessary to highlight thoroughly why it is need to examine the results reported in this article.

Response to comment 2 

We thoroughly revised the introduction section. We toned down any argument not theoretically supported and limited our discussion to that found in the previous literature. Please review our newly revised introduction (p. 2—4).    

Comment 3

H1 is surplus. Many studies had already described this, see e.g. Kluger and DeNisi (1996) or Zhou (1998) in the Journal of Applied Psychology. H2 and H3 make two statements at the same time, which is methodologically incorrect.

Response to comment 3 

We agree that H1 was already tested by much of previous research. However, we need to test it with data from public organizations which were less explored compared to private organizations. The uniqueness of public organizations was described in the introduction. H2 and H3 were revised (p. 10, 12).

Comment 4

As the sample comes from the Korean government, it is necessary to write in detail why these relationships should be investigated in the public sector. What are the characteristics of the Korean organizational cultures in the government? How much feedback is there in relation to the competitive sector or internationally? How are performance expectations handled in government and companies?

Response to comment 4 

The footnote 1 was added to explain the characteristics of the data. Please refer to the footnote 1 on p. 27.

Round 2

Reviewer 1 Report

Dear authors,

Thank you for the revised version of the paper.

I must say that all the requests regarding the previous review report are successfully incorporated into the manuscript.

Well done!

Reviewer 2 Report

Dear Authors,

Thank you for addressing all my concerns. Now the paper is ready to be published.

Best wishes,

Reviewer 3 Report

Thank you for your revision.